# Causal Scaffolding for Physical Reasoning: A Benchmark for Causally-Informed Physical World Understanding in VLMs

## Abstract

Understanding and reasoning about the physical world is the foundation of intelligent behavior, yet state-of-the-art vision-language models (VLMs) still fail at causal physical reasoning, often producing plausible but incorrect answers. To systematically address this gap, we introduce **CausalPhys**, a benchmark of over 3,000 carefully curated video- and image-based questions spanning four domains: Perception, Anticipation, Intervention, and Goal Orientation. Each question is paired with a causal graph that captures underlying interactions and dependencies, enabling fine-grained and interpretable evaluation. We further propose a causal-graph-grounded metric that verifies whether a model's chain-of-thought reasoning follows correct causal relations, moving beyond answer-only accuracy. Systematic evaluations of leading VLMs on CausalPhys expose consistent failures to capture causal dependencies, underscoring fundamental weaknesses in their physical reasoning. To overcome these shortcomings, we introduce a Causal Rationale-informed Fine-Tuning strategy (CRFT) that scaffolds VLM reasoning with causal graphs. Extensive experiments show that CRFT significantly improves both reasoning accuracy and interpretability across multiple backbones. By combining diagnostic evaluation with causality-informed fine-tuning, this work establishes a foundation for advancing VLMs toward causally grounded physical reasoning.

## 1 Introduction

Understanding and reasoning about physical environments is a cornerstone of intelligence, enabling agents to operate robustly in real-world settings (Srivastava et al., 2022; Gupta et al., 2021). Yet today's VLMs remain far from human intuition, often failing to capture even basic physical interactions. Robust physical reasoning demands more than pattern recognition: agents must infer intrinsic object properties (Yi et al., 2019; Chen et al., 2022), track spatial relations across entities (Yang et al., 2025b; Wang et al., 2024), interpret evolving physical scenes, and anticipate how interactions unfold to guide planning and prevent costly errors (Bear et al., 2021; Dong et al., 2025). Humans, by contrast, perform such reasoning effortlessly, drawing on an intuitive grasp of physical causality that emerges early in development (Carey, 2000; McCloskey et al., 1983; Chow et al., 2025). How to equip VLMs with this level of causally grounded understanding remains a central open challenge. Resolving it is critical for advancing embodied AI systems that are both reliable and trustworthy.

Recent VLMs excel at multimodal tasks such as visual question answering, object recognition, and image captioning. Yet extending these successes to dynamic physical reasoning in realistic environments remains an open challenge (Bear et al., 2021; Tung et al., 2023; Chow et al., 2025; Dong et al., 2025). Relying solely on perception-driven capabilities has proven insufficient for building generalist embodied agents (Komanduri et al., 2025; Foss et al., 2025; Liu et al., 2025; Chen et al., 2024), often leading to brittle behaviors such as mishandling fragile objects or misjudging grasp affordances. As a concrete example, Fig. 1 (Intervention) illustrates that inferring the orientation of a door relative to the camera viewpoint from limited observations is far from trivial. Such reasoning demands sensitivity to latent spatial structures, occluded relationships, and viewpoint transformations that are invisible in isolated images. Ultimately, these cases hinge on anticipating how the world changes under interventions or viewpoint shifts, reasoning that is naturally framed through **causal inference**. Structural causal models provide exactly this grounding: they connect inter-

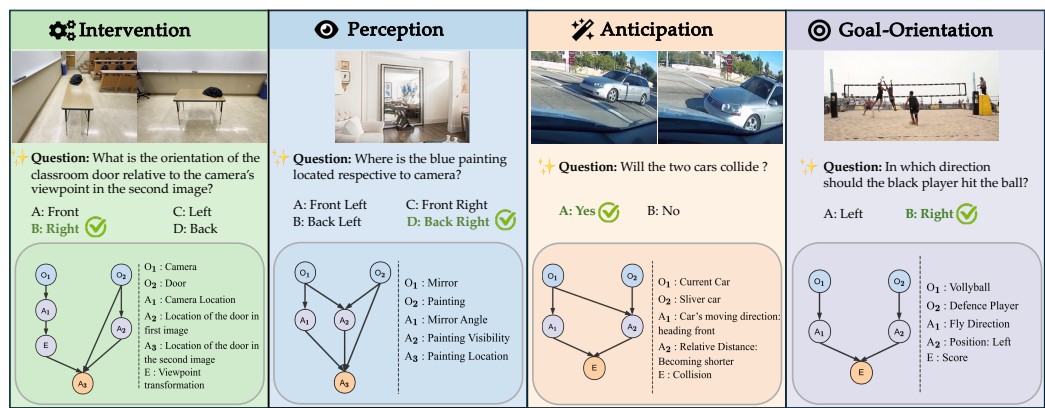

Figure 1: CausalPhys emphasizes comprehensive physical understanding across four distinct categories, with each question explicitly annotated by a causal graph encoding the underlying physical relationships.

ventions with their consequences, enabling agents to infer unobserved properties from incomplete evidence with both precision and interpretability.

However, bringing causal reasoning into VLMs remains non-trivial, and we identify three critical challenges. (1) Current VLMs are trained to capture statistical associations in observational data rather than the underlying causal mechanisms, which limits their ability to reason in dynamic, real-world environments. (2) Existing benchmarks for physical reasoning *rarely include ground-truth causal annotations*, making it impossible to rigorously measure whether models follow correct causal dependencies. (3) There is a notable absence of causally informed fine-tuning methods; prior research efforts in this domain have predominantly focused exclusively on evaluation, leaving a gap in methods that can effectively enhance causal reasoning in multimodal systems. These challenges highlight the absence of targeted training paradigms that explicitly foster causal understanding, underscoring the need for new benchmarks and methods that move VLMs beyond surface correlations toward genuine causal reasoning.

To systematically investigate and address these challenges, we introduce **CausalPhys**, a benchmark of over 3,000 carefully curated video and image questions spanning four domains: Perception, Anticipation, Intervention, and Goal Orientation, across 16 subfields (Fig. 2). Each question is paired with a **ground-truth causal graph** that captures physical interactions and dependencies, enabling **mechanism-level, interpretable evaluation** of VLM reasoning. These explicit annotations make CausalPhys a rigorous testbed for diagnosing both strengths and failures in causal reasoning, filling critical gaps left by prior benchmarks. Using CausalPhys, we systematically evaluate state-of-the-art open-source VLMs and **reveal systematic failures** on tasks that require robust causal inference. Building on these observations, we propose **CRFT**, a causal rationale-enhanced fine-tuning approach that leverages causal graphs to guide VLMs toward generating more accurate and causally consistent explanations, thereby improving both their performance and interpretability in complex physical environments.

We aim for this work to provide meaningful insights and help narrow the gap between VLMs and physical world understanding, thereby fostering progress in embodied AI toward human-level capabilities. By situating our contributions at the intersection of benchmarking, evaluation, and model improvement, we hope to offer a resource that not only diagnoses current limitations but also points toward concrete paths forward. Overall, this paper makes three key contributions. (1) We introduce CausalPhys, the first benchmark to couple real-world physical reasoning questions with explicit ground-truth causal graphs, enabling interpretable, mechanism-level evaluation beyond surface-level answer accuracy. (2) We develop a causal-graph-grounded metric that verifies whether a model's chain-of-thought follows the correct causal dependencies, and we systematically evaluate state-of-the-art VLMs to uncover fine-grained reasoning failures that answer-only benchmarks cannot capture. (3) We propose a causally inspired fine-tuning (CRFT) approach that leverages causal graphs to guide models toward more accurate and consistent reasoning, thereby improving both performance

and interpretability. These contributions push VLMs beyond surface pattern recognition, steering them toward genuine causal reasoning in complex physical environments.

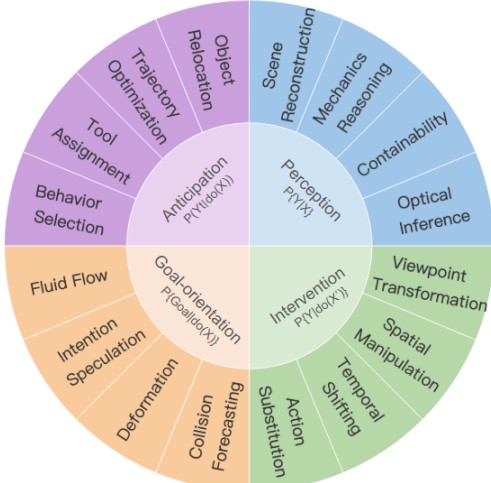

| 👁 **Perception (951)** | | 🏹 **Anticipation (900)** | |
|---|---|---|---|
| **Subset** | **#Question** | **Subset** | **#Question** |
| Optics | 252 | Collision Prediction | 300 |
| Containability | 201 | Deformation | 200 |
| Scene Reconstruction | 200 | Fluid Flow | 200 |
| Mechanics Reasoning | 298 | Intention Speculation | 200 |

| ⚙ **Intervention (573)** | | ◎ **Goal Orientation (638)** | |
|---|---|---|---|
| **Subset** | **#Question** | **Subset** | **#Question** |
| Spatial Manipulation | 151 | Object Relocation | 190 |
| Action Substitution | 99 | Tool Selection | 100 |
| Temporal Shifting | 149 | Behaviour Selection | 229 |
| Viewpoint Transformation | 174 | Trajectory | 119 |

Figure 2: Taxonomy of CausalPhys spanning four causal rungs.

Table 1: Statistics of CausalPhys questions across four domains and 16 subsets.

## 2 RELATED WORKS

**Physical Benchmarks.** Early benchmarks addressing physical reasoning predominantly focused on scenarios involving rudimentary physical interactions and simplified environmental contexts. (Bear et al., 2021; Tung et al., 2023; Zhu et al., 2023) For example, (Yi et al., 2019; Chen et al., 2022)focus on elementary visual primitives, including spheres, cubes, and rigid-body collision events. To assess the physical reasoning capability of VLMs, some datasets (He et al., 2024; Jiang et al., 2024; Lu et al., 2022; Hao et al., 2025; Zhang et al., 2025; Azzolini et al., 2025) are designed, which typically emphasize commonsense reasoning grounded in linguistic knowledge, rather than perceptual understanding of physical interactions. On the other hand, spatial VQA benchmarks (Wang et al., 2024; Yang et al., 2025b; Li et al., 2024; Shiri et al., 2024) focus on geometric relationships and spatial reasoning within 3D scenes, reflecting an initial stage toward comprehensive physical world modeling. Recent advancements include PhysBench (Chow et al., 2025), which has been expanded to provide a comprehensive evaluation of models' understanding of physical scenarios across diverse tasks, and MVPBench (Dong et al., 2025), a curated benchmark specifically developed to assess visual physical reasoning capabilities through visual CoT methodologies. However, these approaches primarily focus on whether VLMs can correctly answer questions, rather than examining the underlying causal reasoning processes, potentially leading to unreliable predictions when applied to real-world environments.

**Causal Reasoning Datasets.** While causal reasoning has been extensively studied for LLMs (Jin et al., 2023; Jiralerspong et al., 2024; Rajendran et al., 2024), equivalent efforts in the VLM domain remain comparatively nascent. Prior work often represents causal structure with narrowly defined nodes. For example, CELLO models nodes as perceptible objects and focuses on simple relations such as "object 1 supports object 2" (Chen et al., 2024). Other approaches seek interpretability by instantiating structural causal models for CoT reasoning (Fu et al., 2025). Recent VLM benchmarks (e.g., CausalVLBench, Causal3D) evaluate causal inference using fixed-structure graphs within predesigned scenes, where models are asked to estimate relations between entities explicitly provided in prompts (Komanduri et al., 2025; Liu et al., 2025). Such constrained experimental setups limit dataset diversity and hinder the capacity of models to uncover generalized causal structures within real-world physical scenarios. These motivate the development of benchmarks that incorporate greater diversity in realistic physical environments, as well as methodologies that explicitly enhance the physical reasoning capabilities of VLMs through causal inference.

Table 2: **Comparison of CausalPhys with existing physical reasoning benchmarks**. While prior datasets are limited by synthetic environments, restricted diversity, or missing causal structure, CausalPhys uniquely integrates **real-world data**, **diverse scenes**, and fine-grained **causal annotations** spanning *objects*, *attributes*, and *events*.

| Dataset | Data Instances | Data Source | | Causal Structure | | Causal Node | | |
| --- | --- | --- | --- | --- | --- | --- | --- | --- |
| | | Real-World Data | Scene Diversity | Annotation | Flexibility | Object | Attribute | Event |
| CELLO (Chen et al., 2024) | 14,000+ | ✓ | ✗ | ✓ | ✓ | ✓ | ✗ | ✗ |
| Causal3D (Liu et al., 2025) | 7 scenes | ✗ | ✗ | ✓ | ✗ | ✗ | ✓ | ✗ |
| CausalVLBench (Komanduri et al., 2025) | 5,000+ | ✗ | ✗ | ✓ | ✗ | ✗ | ✓ | ✗ |
| PhysBench (Chow et al., 2025) | 10,000+ | ✓ | ✓ | ✗ | ✗ | ✗ | ✗ | ✗ |
| Causal VQA (Foss et al., 2025) | 700+ | ✓ | ✓ | ✗ | ✗ | ✗ | ✗ | ✗ |
| MVP Bench (Dong et al., 2025) | 1,000+ | ✓ | ✓ | ✗ | ✗ | ✗ | ✗ | ✗ |
| **CausalPhys (Ours)** | 3,000+ | ✓ | ✓ | ✓ | ✓ | ✓ | ✓ | ✓ |

# 3 THE CAUSALPHYS BENCHMARK

To rigorously assess VLMs' capacity for physical reasoning, we present **CausalPhys**, a benchmark tailored to causally-informed understanding of real-world environments. It comprises over 3,000 image and video instances, each paired with an explicit, instance-specific causal graph, moving beyond fixed causal schemas in prior work. To ensure reproducibility and enable precise capability analysis, we provide a **fully documented data construction pipeline**, along with **open-sourced resources** that make the benchmark both transparent and extensible.

## 3.1 BENCHMARK OVERVIEW

The design of **CausalPhys** builds on a structured taxonomy that aligns physical reasoning tasks with the **rungs of Pearl's causal ladder** (Pearl, 2009). It spans four complementary categories (see Fig. 1): **Perception** (*"What is . . . "*: identifying observable states and attributes), **Anticipation** (*"What will happen next . . . "*: projecting near-future outcomes), **Intervention** (*"What will happen if . . . "*: reasoning about consequences of explicit interventions with the do-operator), and **Goal Orientation** (*"What should be done to achieve . . . "*: planning actions under physical constraints). Each category is further divided into four subcategories, allowing **fine-grained evaluation** of how VLMs reason about different facets of the physical world. By explicitly grounding these categories in the causal hierarchy, CausalPhys provides not only broad coverage of physical reasoning tasks but also a principled framework to reveal *where along the causal rungs VLMs succeed*, and *where they fail*.

Our work bridges a key gap in existing benchmarks, which often reduce causal reasoning to flat graphs with homogeneous node types. Such oversimplification erases the richness of real-world physics, where reasoning must span **objects** with intrinsic attributes, **attributes** that evolve, and **events** that trigger transformations. To capture this heterogeneity, CausalPhys introduces a principled **three-node taxonomy** of *Objects*, *Attributes*, and *Events*, grounding physical reasoning in mechanisms rather than surface correlations. Directed edges encode a wide spectrum of causal dependencies: attributes describing objects, events involving objects, events modifying attributes, and cascades where one event causes another. As illustrated in Fig. 1(Intervention) (under our *viewpoint transformation* category), the event "*Camera rotates counterclockwise by 90°*" alters the positional attribute of the door from *Front to Right*. Each instance is systematically annotated as a **directed acyclic causal graph (DAG)**, making causal dependencies explicit, interpretable, and testable. Our design elevates CausalPhys from a dataset to a diagnostic instrument, revealing not only *what* models predict but also *how* they reason.

Each instance in CausalPhys takes the form of **a multiple-choice question** with two to four options and exactly one correct answer, ensuring unambiguous evaluation. Unlike previous benchmarks designed for physical understanding Dong et al. (2025); Chow et al. (2025), every question in CausalPhys is paired with **a structured causal graph** that encodes the underlying physical mechanisms, making the reasoning process interpretable rather than answer-only. The dataset spans four causal domains and incorporates both images and videos, offering broad coverage of physical reasoning tasks across modalities. Within this setting, causal dependencies are systematically represented: in-

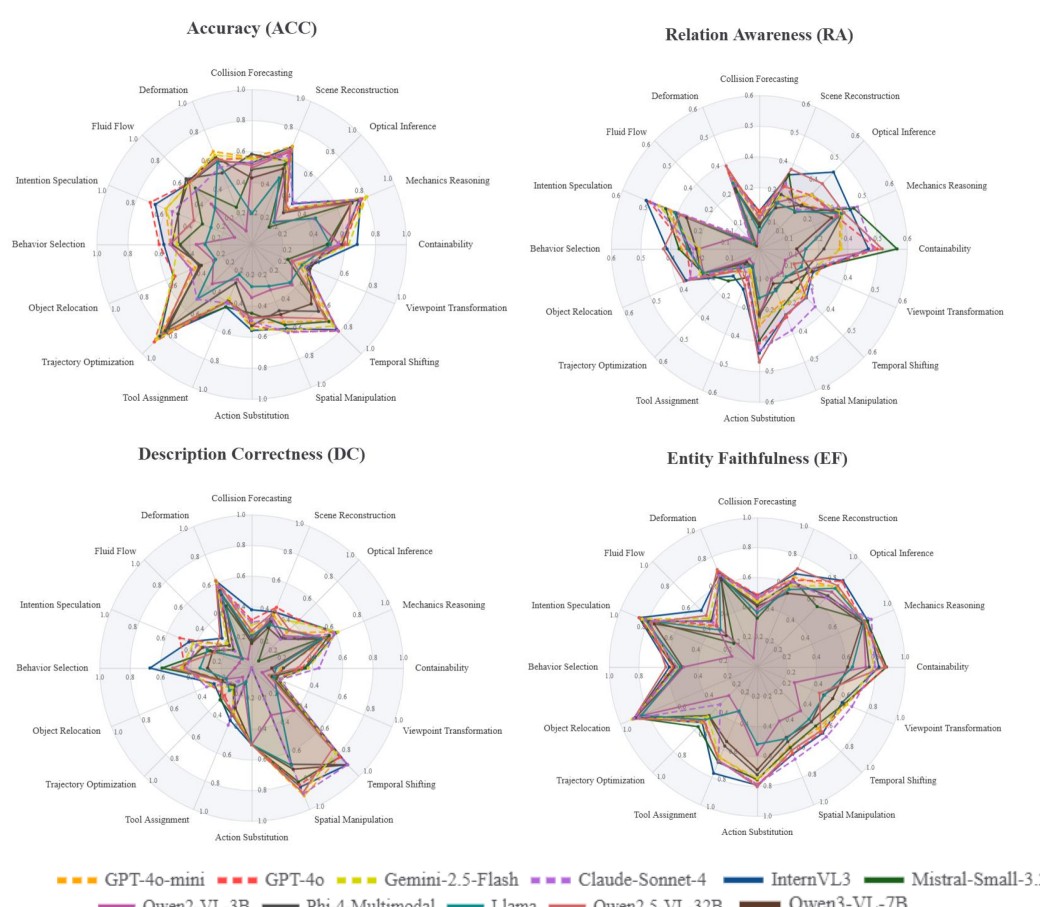

Figure 3: **Radar plots of VLM performance.** Models are evaluated on four metrics: Accuracy (ACC), Relation Awareness (RA), Description Correctness (DC), and Entity Faithfulness (EF) across 16 CausalPhys subcategories.

trinsic object attributes such as velocity or texture, cross-object relations such as relative position, event-driven transformations where collisions alter trajectories, and higher-order chains where one event precipitates another.

## 3.2 DATA COLLECTION WORKFLOW

We built CausalPhys through a carefully structured workflow to ensure both quality and transparency. All images, videos, questions, and annotations were manually curated by annotators with STEM expertise. The construction process followed a structured pipeline of five stages: (a) *Data Acquisition*, where instances were selected from more than ten publicly available datasets with provenance carefully documented; (b) *Question Formulation*, where annotators designed original questions centered on physical commonsense, each explicitly explainable through causal reasoning and paired with verified answers; (c) *Data Processing*, where raw media were standardized and paired with their annotations; (d) *Causal Graph Construction*, where each instance was encoded into a graph, first drafted in Mermaid syntax and then converted into a typed JSON schema with validators to enforce node typing and acyclicity; (e) *Quality Assurance*, where all items underwent double annotation and adjudication to remove cases with insufficient visual cues or textual biases.

To support reproducibility, we release the entire pipeline, including selection scripts, annotation guidelines, Mermaid DAGs encoding causal graphs, JSON schema with validators, and regeneration code for data splits. This design makes CausalPhys not only high-quality but also transparent, auditable, and extensible.

| Model | Open-Source Models | | | | | | | Closed-Source Models | | | |
|---|---|---|---|---|---|---|---|---|---|---|---|
| | InternVL3 (Zhu et al., 2025) | Qwen3-VL (Yang et al., 2025a) | Qwen2.5-VL (Qwen et al., 2025) | Qwen2-VL (Yang et al., 2024) | Llama (Grattafiori et al., 2024) | Phi-4-Multimodal (Abdin et al., 2024) | Mistral-Small-3.2 (Team, 2025) | GPT-4o (Hurst et al., 2024) | GPT-4o-mini (Hurst et al., 2024) | Gemini-2.5-Flash (Comanici et al., 2025) | Claude-Sonnet-4 (Anthropic, 2025) |
| Size | 78B | 32B | 3B | 7B | 11B | 5.6B | 24B | $\sim$200B | 8B | 7B | - |
| 🖋 **Anticipation** | | | | | | | | | | | |
| Accuracy (ACC) ↑ | 0.5800 | 0.5189 | 0.2944 | 0.5222 | 0.3333 | 0.5533 | 0.4100 | **0.6011** | 0.5911 | 0.5822 | 0.5322 |
| Entity Faithfulness (EF) ↑ | **0.6211** | 0.5910 | 0.2700 | 0.5100 | 0.5290 | 0.5570 | 0.4926 | 0.5935 | 0.5706 | 0.5820 | 0.5798 |
| Relation Awareness (RA) ↑ | 0.2338 | 0.2088 | 0.0797 | 0.1710 | 0.1736 | 0.1719 | 0.1808 | **0.2346** | 0.2021 | 0.2061 | 0.2238 |
| Description Correctness (DC) ↑ | **0.4243** | 0.3428 | 0.1217 | 0.2586 | 0.2789 | 0.3012 | 0.2559 | 0.3979 | 0.3303 | 0.3714 | 0.3481 |
| 👁 **Perception** | | | | | | | | | | | |
| Accuracy (ACC) ↑ | **0.6257** | 0.5689 | 0.4490 | 0.5205 | 0.3985 | 0.5573 | 0.4826 | 0.5889 | 0.5983 | 0.5920 | 0.5868 |
| Entity Faithfulness (EF) ↑ | **0.7873** | 0.7822 | 0.7112 | 0.6562 | 0.6965 | 0.7141 | 0.7221 | 0.7738 | 0.7687 | 0.7411 | 0.7349 |
| Relation Awareness (RA) ↑ | **0.3976** | 0.3884 | 0.2692 | 0.2325 | 0.2490 | 0.2822 | 0.3680 | 0.3407 | 0.3027 | 0.3092 | 0.3488 |
| Description Correctness (DC) ↑ | **0.4664** | 0.4049 | 0.3483 | 0.3133 | 0.3457 | 0.3501 | 0.3479 | 0.4621 | 0.4014 | 0.4574 | 0.4218 |
| ⚙ **Intervention** | | | | | | | | | | | |
| Accuracy (ACC) ↑ | **0.5707** | 0.4799 | 0.3246 | 0.4764 | 0.3211 | 0.4852 | 0.4852 | **0.5707** | 0.5131 | 0.5567 | 0.5672 |
| Entity Faithfulness (EF) ↑ | 0.6547 | 0.5941 | 0.3954 | 0.5563 | 0.5044 | 0.5501 | 0.6240 | 0.6592 | 0.6295 | 0.6451 | **0.6941** |
| Relation Awareness (RA) ↑ | 0.2858 | 0.2666 | 0.1493 | 0.2003 | 0.1762 | 0.1925 | 0.2483 | 0.2762 | 0.2496 | 0.2464 | **0.3076** |
| Description Correctness (DC) ↑ | **0.5985** | 0.5516 | 0.2991 | 0.5661 | 0.3781 | 0.4659 | 0.5562 | 0.5742 | 0.5873 | 0.5827 | 0.5278 |
| ◎ **Goal-Orientation** | | | | | | | | | | | |
| Accuracy (ACC) ↑ | 0.5799 | 0.5172 | 0.3103 | 0.4906 | 0.3009 | 0.4483 | 0.4796 | **0.5878** | 0.5157 | 0.5439 | 0.4702 |
| Entity Faithfulness (EF) ↑ | **0.7064** | 0.6978 | 0.5372 | 0.6207 | 0.5762 | 0.6440 | 0.6858 | 0.6784 | 0.6764 | 0.6614 | 0.6470 |
| Relation Awareness (RA) ↑ | 0.2855 | 0.2856 | 0.1819 | 0.1931 | 0.2078 | 0.2036 | 0.2410 | 0.2376 | 0.2238 | 0.1966 | 0.2238 |
| Description Correctness (DC) ↑ | **0.4339** | 0.3564 | 0.1341 | 0.2475 | 0.2264 | 0.3153 | 0.3817 | 0.3447 | 0.3250 | 0.3385 | 0.3439 |
| 📊 **Average** | | | | | | | | | | | |
| Accuracy (ACC) ↑ | **0.5924** | 0.5268 | 0.3514 | 0.5065 | 0.3445 | 0.5199 | 0.4611 | 0.5888 | 0.5630 | 0.5725 | 0.5428 |
| Entity Faithfulness (EF) ↑ | **0.6968** | 0.6795 | 0.4862 | 0.5871 | 0.5862 | 0.6226 | 0.6287 | 0.6795 | 0.6652 | 0.6597 | 0.6634 |
| Relation Awareness (RA) ↑ | **0.3052** | 0.2914 | 0.1729 | 0.2002 | 0.2046 | 0.2166 | 0.2641 | 0.2760 | 0.2468 | 0.2437 | 0.2783 |
| Description Correctness (DC) ↑ | **0.4720** | 0.4040 | 0.2278 | 0.3308 | 0.3073 | 0.3501 | 0.3669 | 0.4397 | 0.3994 | 0.4308 | 0.4037 |

Table 3: **Benchmark evaluation results on CausalPhys.** We report performance of state-of-the-art open- and closed-source VLMs across four domains (Anticipation, Perception, Intervention, and Goal Orientation). Metrics include **Accuracy (ACC)**, **Entity Faithfulness (EF)**, **Relation Awareness (RA)**, and **Description Correctness (DC)**. Results reveal that while models achieve moderate accuracy and entity-level consistency, they struggle with relation-level reasoning (RA), indicating persistent gaps in capturing causal dependencies. These systematic weaknesses underscore the need for causally-informed approaches such as our proposed CRFT.

We further introduce a **causal-graph-grounded evaluation framework** for VLMs which goes beyond answer correctness to assess whether a model's reasoning chain faithfully reflects the underlying physical mechanisms. Given a query $Q$ and an image or video $X$, a vision–language model produces a rationale $R$ and a final answer $Y$. Each instance in CausalPhys is annotated with a ground-truth causal graph $\mathcal{G} = (\mathcal{V}, \mathcal{E}_g)$, where $\mathcal{V}$ contains *objects*, *attributes*, and *events*, and $\mathcal{E}_g$ encodes their directed dependencies. The workflow of the evaluation framework is shown in Fig. 4.

Specifically, our framework evaluates $R$ along four complementary criteria:

1. **Accuracy (ACC).** Measures whether the predicted answer matches the ground-truth label:

$$\text{ACC} = \mathbb{1}\{ Y = Y^* \}.$$

2. **Entity Faithfulness (EF).** Evaluates whether the rationale covers all relevant entities. Let $\mathcal{O}, \mathcal{A}, \mathcal{E}$ denote the sets of objects, attributes, and events in $\mathcal{V}$. For each entity $y \in \mathcal{V}$,

$$\text{EF}(y) = \mathbb{1}\{ y \text{ is explicitly mentioned in } R \}.$$

EF thus measures the coverage of reasoning-relevant entities rather than surface correctness.

3. **Description Correctness (DC).** Checks whether entities are described consistently with their ground-truth annotations. For each $y \in \mathcal{A} \cup \mathcal{E}$ with description $d(y)$,

$$\text{DC}(y) = \mathbb{1}\{ R \text{ contains a description semantically consistent with } d(y) \}.$$

This ensures that models not only mention entities but also characterize them correctly.

4. **Relation Awareness (RA).** Tests whether the rationale captures directed causal dependencies. Let $\mathcal{R}_g$ be the set of directed edges $(u, v)$ in the ground-truth graph, with $u$ as parent of $v$. For each $(u, v) \in \mathcal{R}_g$,

$$\text{RA}(u, v) = \mathbb{1}\{ u, v \in R \ \wedge \ u \text{ is described before } v \}.$$

Here, ordering in $R$ serves as a minimal signal of causal sequencing.

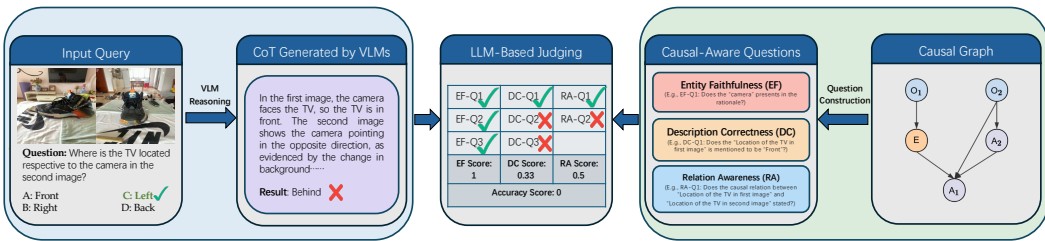

Figure 4: Workflow of evaluating the physical understanding capability of VLMs.

Evaluation proceeds in four stages: (a) **CoT–Answer Generation**, where the VLM outputs $R$ and $Y$; (b) **Answer Verification**, comparing $Y$ against the ground-truth; (c) **Causal-Aware Question Construction**, where auxiliary checks are derived from $\mathcal{G}$ for EF, DC, and RA; and (d) **LLM-based Judging**, which scores the rationale in True/False format and aggregates results into final metrics.

By grounding evaluation in explicit causal graphs, this framework offers fine-grained diagnostics of reasoning, revealing not only whether VLMs are correct but also how their reasoning aligns with the true causal structure of the physical world, and where it diverges.

### 3.3 BENCHMARKING VLMs ON PHYSICAL WORLD UNDERSTANDING

**Understanding physical relations remains challenging for VLMs.**

As shown in Figure 3 and table 3, despite achieving moderate performance on perception-based tasks, current VLMs struggle substantially when reasoning over physical relations. Subsets such as viewpoint transformation remain particularly difficult: most models perform below 40%, essentially at chance. Similarly, in optical inference, where models must predict the true location of an object visible only via a mirror, even state-of-the-art closed-source systems achieve accuracies around 0.3. These tasks require integrating spatial geometry and causal dependencies rather than surface-level cues. While certain subsets, such as trajectory optimization, show surprisingly strong results, these cases are largely driven by perceptual recognition (e.g., detecting a block on a path), not higher-order causal inference. The consistently low performance on relation-intensive subsets highlights the difficulty VLMs face in encoding spatially grounded causal relationships, such as reasoning about visible regions, mirror angles, and relative object positions. These remain open challenges for advancing physical reasoning in complex environments.

**Open-source models perform on par with closed-source systems.**

Unlike many general-purpose multimodal benchmarks, our results reveal that open-source models match the performance of proprietary systems. The average gap between open and closed source models is negligible across categories. For example, InternVL3 achieves nearly the same overall accuracy as GPT-4o while maintaining consistently high entity faithfulness. In anticipation and intervention, the performance difference between open and closed models is especially narrow, indicating that scale or private data access is not the sole determinant of success in physical reasoning. This suggests that strong open-source efforts provide credible baselines for causal reasoning tasks, reducing dependence on closed systems.

Beyond the open–closed distinction, differences across model sizes emerge as more pronounced. Within the Qwen series, for instance, smaller variants (e.g., Qwen2-VL 3B) perform substantially worse than their larger counterparts (7B and 32B) across all four categories. The 3B model often lags by wide margins in both accuracy and causal relation awareness, suggesting that limited capacity constrains its ability to perform structured reasoning. By contrast, the 7B and 32B models demonstrate clear gains, indicating that sufficient scale is critical for capturing the causal dependencies required by our benchmark.

**Causal relation awareness is most predictive of reasoning success.**

While entity faithfulness (EF) is generally high and the rate often exceeding 0.65, model accuracy correlates more strongly with relation awareness (RA). Models that capture causal links within the ground-truth graph tend to achieve higher accuracies, even when EF remains broadly similar across

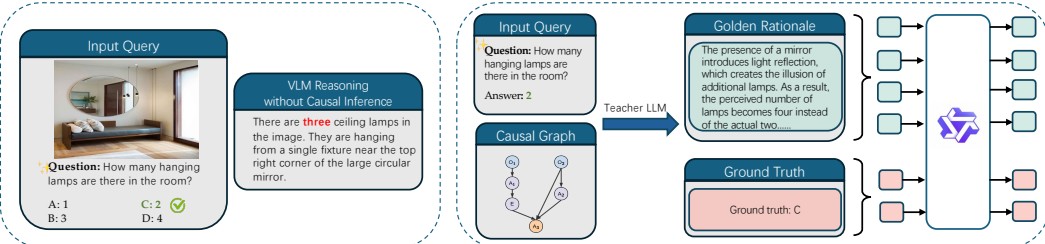

Figure 5: The workflow of CRFT. CRFT employs causal graphs to guide VLMs in generating more precise, causally coherent physical reasoning. The causal rationale is jointly optimized with the ground-truth answer.

systems. For example, in perception and intervention tasks, improvements in RA are closely aligned with accuracy gains, whereas EF shows weaker variance. This suggests that correctly identifying entities alone is insufficient for reasoning: assembling them into coherent causal structures is critical. Relation awareness thus emerges as the strongest correlate of final task performance, highlighting it as a key bottleneck for progress.

**A persistent gap exists between entity recognition and relation reasoning.**

Across all categories, we observe a consistent EF–RA gap: models readily identify relevant objects and attributes but struggle to connect them through causal relations. For instance, EF scores remain high ($\approx 0.7$), while RA lags behind ($\approx 0.2$–$0.3$). This indicates that models succeed at local recognition yet fail at deeper relational inference. Bridging this gap appears essential for advancing beyond surface-level performance. Closing the EF–RA gap may therefore be the critical step toward robust causal reasoning in vision-language models.

## 4 FROM ANSWERS TO REASONS: CAUSAL RATIONALE FINE-TUNING

Our benchmark analysis (Sec. 3.3) reveals a consistent pattern: VLMs perform better when they articulate not only the final answer, but also the causal relations that explain it. Motivated by this insight, we propose **Causal Rationale Fine-Tuning (CRFT)**, a training paradigm that explicitly grounds rationales in causal graphs, teaching VLMs to reason through mechanisms rather than surface correlations. The workflow of CRFT is shown in Fig. 5.

**Rationale Construction.** Given a dataset of instances $(x, q, y, G)$, where $x$ is an image or video, $q$ a query, $y$ the correct answer, and $G = (\mathcal{N}, \mathcal{E})$ a human-annotated causal graph, we generate **gold causal rationales** $r$ using a teacher LLM (e.g., GPT-4o (Hurst et al., 2024)). Each rationale is required to (i) explicitly reference nodes and edges in $G$, (ii) trace intermediate causal implications, and (iii) conclude with $y$. This ensures that rationales are *faithful to the causal graph*, providing structured supervision beyond free-form text.

**Training Objective.** For training, we concatenate the rationale and the answer into a single sequence $s = [r; y]$ and fine-tune the target VLM $\pi_\theta$ to maximize its likelihood under a weighted supervision scheme:

$$\mathcal{L}_{\text{CRFT}}(\theta) = -\mathbb{E}_{(x,q,y,G)\sim\mathcal{D}}\left[\lambda_r \sum_{t\in\text{idx}(r)} \log \pi_\theta(s_t|x, q, s_{<t}) + \lambda_y \sum_{t\in\text{idx}(y)} \log \pi_\theta(s_t|x, q, s_{<t})\right], \quad (1)$$

where $\lambda_r$ and $\lambda_y$ balance rationale and answer supervision.

By anchoring fine-tuning to causal rationales, CRFT compels VLMs to **internalize causal pathways** rather than memorize surface correlations. The model is guided not just to deliver the right answer, but to trace *why* the answer follows, aligning its reasoning with the ground-truth causal graph. This shift from *answers to reasons* transforms evaluation into learning: it produces predictions that are more accurate, reasoning that is more interpretable, and models that are ultimately more reliable

| Metric | Models | | |
|---|---|---|---|
| | QWEN2-VL 7B | QWEN2-VL 7B SFT | QWEN2-VL 7B CRFT |
| Accuracy (ACC) ↑ | 0.5349 | 0.6762 | **0.7066** |
| Entity Faithfulness (EF) ↑ | **0.5978** | 0.3247 | 0.5969 |
| Relation Awareness (RA) ↑ | 0.2130 | 0.0911 | **0.2554** |
| Description Correctness (DC) ↑ | 0.2905 | 0.2645 | **0.3493** |

Table 4: **Average results of QWEN2-VL 7B models on CausalPhys.** We report mean performance across all tasks for the vanilla, SFT, and CRFT checkpoint variants. Best values are in bold.

for physical decision-making. Such alignment is uniquely enabled by CausalPhys, where every instance comes with explicit causal structure and gold rationales, making CRFT both principled and practically feasible.

As shown in Table 4, observed from the fine-tuning results, we can conclude that although SFT (answer-only fine-tuning) achieves a satisfactory Accuracy (ACC) score, its performance on Entity Faithfulness (EF), Description Correctness (DC), and especially Relation Awareness (RA) drops dramatically. This suggests that answer-only supervision encourages the model to optimize for surface-level prediction accuracy, but at the cost of its ability to capture and reflect the underlying causal reasoning process. In other words, SFT fine-tuning tends to make the model behave like a "guesser," prioritizing conclude final answers based on shallow experience rather than over structured, interpretable reasoning chains.

In contrast, the proposed CRFT introduces causal relations explicitly into the fine-tuning strategy. The results show that CRFT not only preserves competitive accuracy but also substantially improves EF, DC, and RA scores compared to both the vanilla model and the SFT-only variant. This indicates that CRFT encourages the model to ground its answers in a more faithful and structured causal rationale, aligning outputs more closely with human-like reasoning. Importantly, CRFT demonstrates that integrating causal structure into fine-tuning can mitigate the trade-off between accuracy and interpretability, producing models that are both effective in prediction and transparent in reasoning.

## 5 CONCLUSION

In this paper, we introduced CausalPhys, a comprehensive benchmark that grounds physical reasoning evaluation in explicit causal graph annotations. By moving beyond surface-level question answering, CausalPhys provides a structured and principled way to dissect the reasoning capabilities of VLMs across perception, anticipation, intervention, and goal-oriented tasks. Our extensive experiments reveal that even the strongest state-of-the-art VLMs struggle when reasoning requires causal consistency, highlighting a fundamental gap between pattern recognition and true physical understanding. To bridge this gap, we proposed Causal Rationale Fine-Tuning (CRFT), which injects causal structure into model training, enabling VLMs to generate answers supported by faithful, interpretable reasoning chains.

Looking ahead, CausalPhys opens several avenues for advancing causal physical reasoning in AI. First, the benchmark can be extended to richer physical scenarios involving stochastic dynamics, long-horizon dependencies, and multi-agent interactions, thereby pushing models closer to real-world complexity. Second, incorporating embodied simulations, where agents not only observe but also act upon environments, will allow us to assess whether VLMs can transfer causal reasoning into interactive decision-making. Third, future work could explore causal generalization, evaluating whether models trained on one set of causal structures extrapolate to novel but related ones, a hallmark of robust reasoning. Finally, the integration of causal rationales with reinforcement learning and embodied robotics offers an exciting path toward building AI systems that are not only accurate but also trustworthy, interpretable, and capable of reasoning like scientists about the physical world.

ETHICS STATEMENT

This work adheres to the ICLR Code of Ethics. Our benchmark, CausalPhys, is constructed entirely from existing, publicly available datasets, including Ego4D, Epic Kitchen, SportsMOT, Something-Something, MSD, CausalVQA, Fluid Flow, Cup dataset, Nexar Collision Prediction, MindCube, and Video Dataset of Human Demonstrations. These sources were selected for their focus on physical interactions and causal dynamics, and all are already distributed for research purposes. No personally identifiable or sensitive data is included in CausalPhys.

By curating from synthetic and publicly released visual data, we minimize risks related to privacy, fairness, or real-world safety. Nonetheless, we acknowledge that improvements in causal reasoning for vision–language models could have downstream societal implications if applied in sensitive domains such as robotics, surveillance, or industrial automation. Our contributions are intended solely for advancing scientific research in multimodal reasoning and should be carefully evaluated before deployment in real-world systems. We declare no conflicts of interest or external sponsorship influencing this work.

REPRODUCIBILITY STATEMENT

We have taken deliberate steps to ensure reproducibility of our results. The CausalPhys benchmark, along with data splits and annotations, is publicly available. Detailed descriptions of dataset construction, evaluation metrics, and experimental setups are provided in Section **??**, with implementation details and prompt templates included in Appendix **??**. Anonymized source code, configuration files, and evaluation scripts are provided with the submission to enable independent verification and replication of all reported results.

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

# APPENDIX

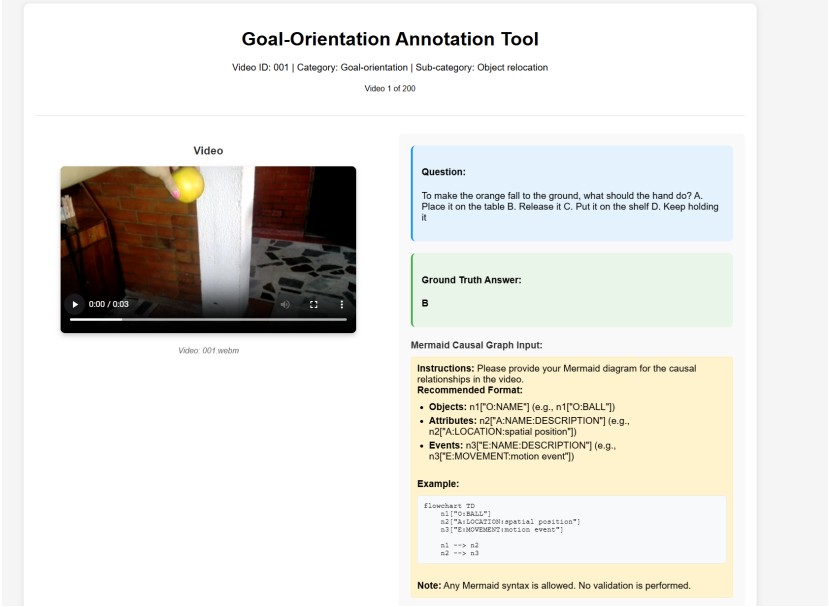

Figure 6: The annotation tool (GUI): Data presentation and question annotation

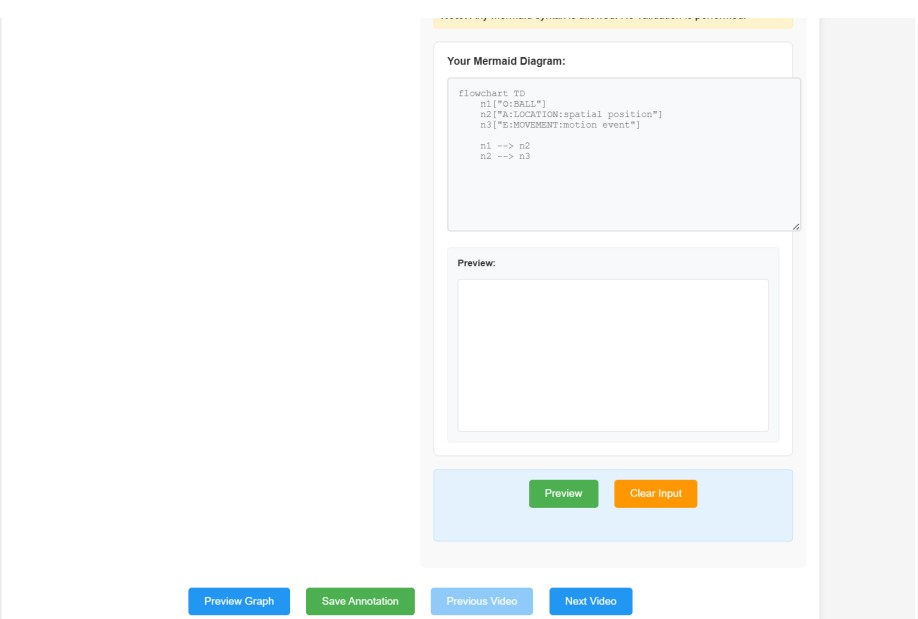

Figure 7: The annotation tool (GUI): Causal graph annotation

## .1 PROMPTS

**Gold Rationale generation prompt:**

You are a reasoning assistant that analyzes visual scenarios and provides step-by-step reasoning.

**INPUT FORMAT:**

You will receive:
- A question about the visual scenario
- A ground truth answer (A, B, C, or D)
- Supporting information about objects, their properties, and relationships

**TASK:**

Generate a clear, step-by-step rationale that answers the question in natural language.

**REQUIREMENTS:**

1. Write an objective, answer-focused rationale in natural language
2. Treat the supporting information as reference only (do not describe it)
3. Write ONE coherent paragraph (max 8 sentences) that flows naturally
4. Include relevant elements from the reference only when needed for reasoning (do not enumerate them)
5. Follow the correct logical order: causes must appear before their effects in your explanation
6. If an element has a description, you MUST state it clearly and exactly as provided
7. Use natural, everyday language (avoid terms like "entity", "relation", "graph", "structure")
8. Ensure proper grammar and spelling
9. Make the explanation easy to understand and self-contained
10. Present the reasoning as a logical analysis of the situation

**OUTPUT FORMAT:**

- Single paragraph only
- No bullet points, lists, or special formatting
- Plain English text written
- Complete explanation that follows the logical reasoning sequence

**IMPORTANT:**

The supporting information (entities, descriptions, relations) is for reference only. Do NOT describe or list it. Use it implicitly to justify the answer. Focus on explaining why the answer is correct in plain language.

**VLM reasoning prompt with rationale**

You are a precise Vision–Language QA assistant.

**GOALS**

Read the user's question and (if provided) a **SEQUENCE** of images in the given order.
Provide a one-sentence rationale and your answer.

**SEQUENCE HANDLING**

If multiple images are provided, treat them as an ordered sequence (e.g., frames of a video).
Consider temporal consistency and cross-frame cues when reasoning.

**CONSERVATIVE REASONING**

Rely only on information available in the images and the question.
Be explicit and concise; avoid speculation.

**HARD FORMAT CONSTRAINTS (MUST OBEY EXACTLY)**

Output MUST include:
1. Generate a clear, step-by-step rationale (max 8 sentences) wrapped in `<rationale>...</rationale>` tags
2. Your answer must be in **EXACTLY ONE CAPITAL LETTER: A, B, C, or D** wrapped in `<result>...</result>` tags

**VLM reasoning prompt answer only**

You are a precise Vision–Language QA assistant.

**GOALS**

Read the user's question and (if provided) a **SEQUENCE** of images in the given order.
Answer with ONLY a single capital letter: A, B, C, or D.

**SEQUENCE HANDLING**

If multiple images are provided, treat them as an ordered sequence (e.g., frames of a video).
Consider temporal consistency and cross-frame cues when reasoning.

**CONSERVATIVE REASONING**

Rely only on information available in the images and the question.
Be explicit and concise; avoid speculation.

**HARD FORMAT CONSTRAINTS (MUST OBEY EXACTLY)**

Output MUST consist of **EXACTLY ONE CAPITAL LETTER: A, B, C, or D.**

**LLM as a judge causal relationship prompt**

You are a meticulous evaluator. Read the problem and the model's rationale, then answer a list of True/False questions strictly based on that rationale. Do not use outside knowledge or the image. If the rationale is ambiguous or does not state the fact, answer **False**.

**Answer using ONLY the specified YAML schema. Do not add extra commentary.**

**INPUT**

- problem: The multiple-choice question with options
- rationale: The model's rationale paragraph(s)
- questions: A list of True/False questions. Each item has:
- id: opaque identifier (string)
- text: the T/F question

**JUDGING PRINCIPLES**

- True only if the rationale explicitly supports the statement with clear mention or an unambiguous entailment.
- False if absent, unclear, contradicted, or only weakly implied.
- Allow synonyms/coreference (e.g., "kicker" for "Fighter"), but do not infer beyond text.
- For causal relation questions, require a clear causal/influence expression (e.g., X causes/leads to/affects Y; Y depends on X). Mere co-occurrence is insufficient.

**OUTPUT FORMAT (YAML)**

```
answers:
- id:  <string>
answer:  true|false
```

**EXAMPLE**

```
problem:  "Which direction should he kick to hit the target?
A. Left B. Right"
rationale:  "The pad is to the left of the kicker; therefore he
should kick left to hit it."
questions:
- id:  "0"
text:  "Is object 'Fighter' mentioned in the rationale?"
- id:  "1"
text:  "Is the causal relation between 'Kick direction' and
'Fighter' correctly expressed?"

answers:
- id:  "0"
answer:  true
- id:  "1"
answer:  true
```

.2 CASE STUDY

Anticipation: Collision Prediction

**Question 1:**

Based on the current frames, will the car collide?

**Option:**

A. Yes

B. No

**Model Response:** B—The sequence of images shows a car following another vehicle on a highway. The distance between the two cars remains consistent across the frames, indicating that the following car is maintaining a safe distance. There are no sudden changes in speed or direction visible in the frames. The road conditions appear stable, and there are no visible obstacles or sudden stops by the leading car. The traffic flow seems steady, with no indications of abrupt braking or lane changes. Therefore, based on the current frames, a collision is unlikely.

**Ground Truth**: A          **Mermaid Flow:**

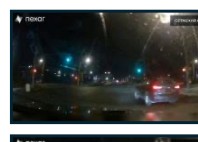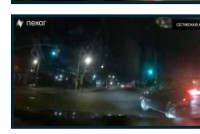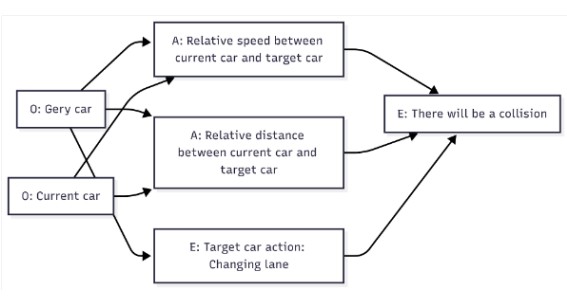

**Question 2:**

Based on the current frames, will the car collide?

**Option:**

A. Yes

B. No

**Model Response :** B—In the sequence of images, the white car on the right is moving into the lane of the black car in front. The white car's position changes slightly towards the black car, indicating a lane change. However, the distance between the two cars remains consistent, suggesting that the white car is not moving fast enough to cause a collision. The black car appears to maintain its speed and position, indicating no immediate threat of collision. The traffic is moving slowly, which reduces the likelihood of a collision

**Ground Truth**: A          **Mermaid Flow:**

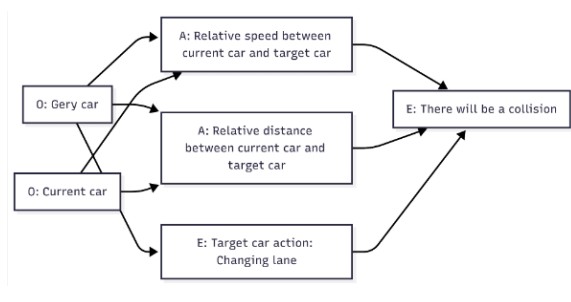

Figure 8: Sample cases of Anticipation-Collision Prediction

## USE OF LARGE LANGUAGE MODELS

We used large language models solely as auxiliary tools to assist in the language polishing and stylistic refinement of the paper draft. The research ideas, methodology, experiments, and analysis

## Anticipation: Deformation

**Question 1:**

What will happen to the garment as the person continues their current action?

**Option:**

A. The garment will be smoothed out completely      B. The garment will be wrinkled further

C. The garment will be folded in half      D. The garment will be stretched out

**Model Response:** C--The images show a person folding a garment on a table. The person's hands are positioned on the garment, indicating they are in the process of folding it. The garment is already partially folded, and the person's hands are moving in a way that suggests they are continuing to fold it. There are no signs of stretching or wrinkling actions being performed. The setup and positioning of the hands imply that the garment will be folded further rather than smoothed out or stretched.

**Ground Truth**: A      **Mermaid Flow:**

**Question 2:**

What will happen to the blue T-shirt as the person continues their current action?

**Option:**

A. It will become smoother.      B. It will get wrinkled.

C. It will tear.      D. It will change color.

**Model Response:** A--The images show a person folding a blue T-shirt on a table. The person is using their hand to smooth out the T-shirt, which is a common action to remove wrinkles and make the fabric lie flat. The T-shirt is already partially folded and appears to be in the process of being smoothed out. There is no indication of any action that would cause the T-shirt to tear or change color. Therefore, as the person continues their current action, the T-shirt will likely become smoother.

**Ground Truth**: A      **Mermaid Flow:**

Figure 9: Sample cases of Anticipation-Deformation

**Anticipation: Fluid Flow**

**Question 1:**

Where will the liquid from container flow?

**Option:**

A. Into the other container    B. No outflow

C. Onto the table                    D. Onto the hand

**Model Response:** B--The sequence of images shows a hand reaching for a plastic cup filled with liquid. The cup is being lifted, but there is no indication of the liquid being poured or spilled. The other container is a paper cup, which remains stationary and empty throughout the sequence. The hand is holding the cup steadily, suggesting no liquid is flowing out. There is no visible movement of liquid towards the table or the hand. Therefore, the liquid remains in the cup without any outflow.

**Ground Truth**: A          **Mermaid Flow:**

**Question 2:**

Where is the liquid from container expected to go?

**Option:**

A. No outflow                      B. Onto the table

C. Onto the robotic arm      D. Into the other container.

**Model Response:** D--The sequence of images shows a robotic arm holding a container with liquid. The arm is tilting the container towards a glass on the table. The position and angle of the container suggest that the liquid is being poured into the glass.The glass is positioned directly under the spout of the container, indicating the intended target for the liquid. There is no indication of the liquid spilling onto the table or the robotic arm.

**Ground Truth**: B          **Mermaid Flow:**

Figure 10: Sample cases of Anticipation-Fluid Flow

Anticipation: Intention Speculation

**Question 1:**

What will the person's next move be?

**Option:**

A. Pick up cafetiere plunger                    B. Pick up cutlery

C. Close cupboard                               D. Put plate into cupboard

**Model Response:** D--The sequence of images shows a person in a kitchen. In the first image, they are near a sink with a drying rack. In the second image, they are holding a clean plate. In the third image, they open a cupboard containing plates. In the fourth image, they are holding the plate near the cupboard. In the fifth image, they are about to place the plate into the cupboard. The logical next step is to put the plate into the cupboard.

**Ground Truth**: C          **Mermaid Flow:**

**Question 2:**

What step will the person take next?

**Option:**

A. Open fridge                                  B. Close fridge

C. Move bottle to the second hand               D. Pick up towel

**Model Response:** A--In the first image, the person is holding a blue container near the sink. In the subsequent images, the person is moving towards the fridge while still holding the container. The final image shows the person standing directly in front of the fridge, suggesting the next logical step is to open it.

**Ground Truth**: A          **Mermaid Flow:**

Figure 11: Sample cases of Anticipation-Intention Speculation

Goal Orientation : Object Relocation

**Question 1:**

To cover the bowl of flour with the lid, in which direction should the hand move?

**Option:**

A. Right          B. Left

C. Forward        D. Upward

**Model Response:** B-- The image shows a bowl with a lid placed to the right of it. To cover the bowl with the lid, the hand should move the lid towards the bowl. Since the lid is on the right side of the bowl, the hand should move left to place the lid over the bowl.

**Ground Truth:** B          **Mermaid Flow:**

**Question 2:**

To place the blue cup into the pink basket, in which direction should the hand move?

**Option:**

A. Right          B. Forward

C. Left           D. Backward

**Model Response:** B--The image shows a hand holding a blue cup near a pink basket. The basket is positioned with its opening facing the hand. To place the cup into the basket, the hand needs to move towards the basket's opening. The basket is directly in front of the hand, so the hand should move forward. The direction \"forward\" aligns with moving towards the basket's opening.

**Ground Truth:** A          **Mermaid Flow:**

Figure 12: Sample cases of Goal Orientation-Object Relocation

Goal Orientation: Tool Selection

**Question 1:**

Which tool will be used to install the Rashult assemble bracket by tightening the screws?

**Option:**

A. Phillips screwdriver          B. Flathead screwdriver

C. Allen key                          D. Wrench

**Model Response:** A--The image shows a close-up of a hand holding a bracket with visible screws. The screws have a cross-shaped indentation, which is characteristic of Phillips screws. This indicates that a Phillips screwdriver is needed to tighten them There are no hexagonal or flathead features visible that would suggest the use of an Allen key, flathead screwdriver, or wrench.

**Ground Truth**: A          **Mermaid Flow:**

**Question 2:**

Which tool will be used to disassemble the MARIUS stool by loosening its screws?

**Option:**

A. Allen key with handle (looped hex key)          B. Phillips screwdriver

C. Flathead screwdriver                                        D. Wrench

**Model Response:** D--The image shows a person using a wrench to loosen a bolt on the underside of a stool. The bolt is hexagonal, which is typically loosened with a wrench. There is no indication of a hex key, Phillips, or flathead screwdriver being used. The presence of a wrench in the person's hand confirms that this is the tool being used for disassembly.

**Ground Truth**: A          **Mermaid Flow:**

Figure 13: Sample cases of Goal Orientation-Tool Selection

1242
1243
1244
1245
1246
1247
1248
1249
1250
1251
1252
1253
1254
1255
1256
1257
1258
1259
1260
1261
1262
1263
1264
1265
1266
1267
1268
1269
1270
1271
1272
1273
1274
1275
1276
1277
1278
1279
1280
1281
1282
1283
1284
1285
1286
1287
1288
1289
1290
1291
1292
1293
1294
1295

## Goal Orientation: Behavior Selection

**Question 1:**

Which direction should he move to get past the defender in soccer ?

**Option:**

A. Left          B. Right

**Model Response:** B--The image shows a soccer player in a white jersey attempting to maneuver past a defender in a green and black jersey. The player in white is using his right foot to control the ball, and the defender is positioned directly in front of him. The defender's stance and position suggest that moving to the left would be more challenging due to the defender's proximity. Moving to the right appears to be a more open path, as the defender's body is angled slightly away from that direction, potentially allowing the player in white to bypass the defender more easily.

**Ground Truth**: B          **Mermaid Flow:**

**Question 2:**

Which direction should the teammate in red clothing move to position themselves properly during the offensive and defensive confrontation?

**Option:**

A. Left          B. Right

**Model Response:** B--The image shows a player in red attempting to tackle a player in white and blue. The player in red is bent over, indicating they are engaged in a defensive action. To position themselves properly, the teammate in red should move to the right to support the tackle and cover any potential escape routes for the player in white and blue.

**Ground Truth**: B          **Mermaid Flow:**

Figure 14: Sample cases of Goal Orientation-Behavior Selection

## Goal Orientation: Trajectory

**Question 1:**

To score and hit the left side of the rim, in which direction should the shot be shifted?

**Option:**

A. Up          B. Down

C. Left          D. Right

**Model Response:** C--The sequence of images shows a basketball being shot towards the hoop. In the final frames, the ball is approaching the center of the hoop. To hit the left side of the rim, the shot needs to be adjusted horizontally. Since the ball is currently centered, shifting the shot to the left would make it hit the left side of the rim.

**Ground Truth:** C          **Mermaid Flow:**

**Question 2:**

To score, in which direction should the shot be adjusted?

**Option:**

A. Up          B. Down

C. Left          D. Right

**Model Response:** C--In the sequence of images, the basketball is seen approaching the hoop. In the final frames, the ball is slightly to the right of the hoop. This indicates that the shot is likely to miss on the right side. Therefore, to increase the chances of scoring, the shot should be adjusted to the left.

**Ground Truth:** D          **Mermaid Flow:**

Figure 15: Sample cases of Goal Orientation-Trajectory

**Intervention: Spatial Manipulation**

**Question 1:**

If the player wearing blue stands 2 meters away from the player in yellow instead of standing right in front of her, will he still be able to intercept the ball?

**Option:**

A. Yes.          B. No.

**Model Response:** B—In the sequence of images, the player in blue is closely guarding the player in yellow, who is attempting to pass the ball. The blue player's proximity allows for a successful interception attempt. If the blue player were to stand 2 meters away, the increased distance would reduce the likelihood of intercepting the ball due to the decreased ability to react quickly to the pass. The close proximity is crucial for the interception, as it allows the blue player to immediately contest the pass.

**Ground Truth**: B          **Mermaid Flow:**

**Question 2:**

If the player in white jumps 0.2 meters higher when he tries to contest for the possession of the ball, will he successfully touch the basketball ?

**Option:**

A. Yes          B. No

**Model Response:** A—In the sequence of images, the player in white is attempting to jump for the basketball during the tip-off. In the final image, the player in white is very close to the ball but does not appear to touch it. If he jumps 0.2 meters higher, it is likely that he will successfully reach and touch the basketball, as the additional height would compensate for the small gap observed.

**Ground Truth**: A          **Mermaid Flow:**

Figure 16: Sample cases of Intervention-Spatial Manipulation

**Intervention: Action Substitution**

**Question 1:**

Based on the position and motion of both the offensive and defensive players, would the shooter have a higher or lower chance of being blocked if she jumps forward while taking the shot? **Option:**

A. higher          B. lower

**Model Response:** A--In the first image, the offensive player in blue is preparing to shoot with a defender in yellow closely positioned in front. In the second image, the shooter is starting to jump, and the defender is also preparing to jump. In the third image, the defender is in the air, closely contesting the shot. In the fourth image, the defender's hand is near the ball, indicating a strong contest. In the fifth image, the defender's hand is very close to the ball, suggesting a high chance of blocking. The shooter's forward motion would bring her closer to the defender, increasing the likelihood of a block.

**Ground Truth**: A          **Mermaid Flow:**

**Question 2:**

Is it likely that the player in yellow could still receive the ball if it were passed 50 cm more to the right from the current trajectory?

**Option:**

A. Yes          B. No

**Model Response:** A--In the sequence of images, the player in yellow is running towards the basket while the player with the ball is also moving in the same direction. The ball is currently being dribbled by the player in yellow, indicating control over the ball. The player in yellow is ahead of the defenders and has a clear path. If the ball were passed 50 cm more to the right, it would still be within reach of the player in yellow, given their proximity and forward momentum. The player in yellow appears to be in a position to adjust and receive the ball even if it deviates slightly from the current trajectory.

**Ground Truth**: A          **Mermaid Flow:**

Figure 17: Sample cases of Intervention-Action Substitution

## Intervention: Temporal Shifting

**Question 1:**

If the shooter delays the shot by 1 second while staying in the air, what will happen?

**Option:**

A. The ball is blocked by the defender.       B. The shot still goes cleanly toward the basket.

C. The player lands before releasing, causing a turnover.

**Model Response:** C--The image shows a basketball player in mid-air attempting a shot with a defender closely contesting. The defender's arm is extended, and they are in a position to potentially block the shot. If the shooter delays the shot by 1 second while staying in the air, it is likely that the defender will have more time to react and block the shot. Additionally, staying in the air for an extra second is unrealistic due to gravity, suggesting the player would land before releasing the ball, causing a turnover.

**Ground Truth**: A          **Mermaid Flow:**

**Question 2:**

If player 14 waits 2 seconds longer to catch the ball, will he still successfully receive the ball?

**Option:**

A. No.     B. Yes.

**Model Response:** A--In the first image, player 14 is positioned to receive the ball. In the second image, the ball is in the air, moving towards player 14. By the third image, player 14 is closer to the ball, indicating it is still on its way. In the fourth image, the ball is nearing player 14, but other players are also approaching. By the fifth image, player 14 is about to receive the ball, but opponents are closing in. If player 14 waits 2 seconds longer, the opponents will likely intercept the ball before it reaches him.

**Ground Truth**: A          **Mermaid Flow:**

Figure 18: Sample cases of Intervention-Temporal Shifting

**Intervention: Viewpoint Transformation**

**Question 1:**

Where is the red pipeline in the first image located relative to the camera view in the second image ?

**Option:**

A. front          B. left

C. below

**Model Response:** B--In the first image, the red pipeline is visible on the wall above the table and near the door. In the second image, the camera is positioned further down the hallway, looking back towards the area where the table and yellow boxes are located. The red pipeline is not directly visible in the second image, but based on the perspective, it would be located to the left of the camera view, as the camera is facing down the hallway away from the wall where the pipeline is mounted.

**Ground Truth**: A          **Mermaid Flow:**

**Question 2:**

Where is the ottoman in the first image located relative to the camera view in the second image? **Option:**

A. right          B. back

C. front          D. left

**Model Response:** D--In the first image, the ottoman is positioned to the right of the green recycling bin. In the second image, the camera is facing the bins from the side, with the green bin visible behind the blue bin. Since the ottoman is to the right of the bins in the first image, and the camera in the second image is facing the bins from the side, the ottoman would be located to the left of the camera's view in the second image.

**Ground Truth**: C          **Mermaid Flow:**

Figure 19: Sample cases of Intervention-Viewpoint Transformation

## Perception: Optics

**Question 1:**

Where is the object chair located with respect to camera's view?

**Option:**

A. Front Left          B. Front Right

C. Back Left          D. Back Right

**Model Response:** D--The image shows a book in front of a mirror, with a chair visible in the reflection. The chair is located in the background, to the right side of the image. Since the chair is in the reflection, it is positioned behind the camera's view. Therefore, the chair is located at the back right with respect to the camera's view.

**Ground Truth**: D          **Mermaid Flow:**

**Question 2:**

Where is the object box located with respect to camera's view?

**Option:**

A. Front Left          B. Front Right

C. Back Left          D. Back Right

**Model Response:** A--The image shows a box with a mouse graphic and text on it. The box is positioned towards the left side of the image. The camera's view captures the box in the foreground, indicating it is closer to the camera. The orientation and placement suggest that the box is located in the front left of the camera's view.

**Ground Truth**: C          **Mermaid Flow:**

Figure 20: Sample cases of Perception-Optics

Perception: Containability

**Question 1:**

If we have two identical glasses vases with same shape, like the one shown in the image, can they be nested into each other properly?

**Option:**

A. Yes.          B. No.

**Model Response:** A--The image shows a glass vase with a wide opening and a narrow base. For two identical vases to be nested, the opening must be wide enough to accommodate the base of the other vase. Since the vase has a narrow base and a wide opening, it is likely that they can be nested into each other properly.

**Ground Truth:** B          **Mermaid Flow:**

**Question 2:**

If we have two identical cups with same shape, like the one shown in the image, can they be nested into each other properly?

**Option:**

A. Yes          B. No

**Model Response:** A--The image shows a glass with a Bloody Mary and a celery stick. The glass appears to have straight sides and a uniform shape, which typically allows for nesting. Glasses with straight sides and no protrusions or handles can usually be nested into each other properly. Therefore, two identical cups with the same shape as the one shown should be able to nest into each other.

**Ground Truth:** A          **Mermaid Flow:**

Figure 21: Sample cases of Perception-Containability

## Perception: Scene Reconstruction

**Question 1:**

Will the ball cross the sideline?

**Option:**

A. Yes          B. No

**Model Response:** A--The image shows a soccer match with the ball in mid-air. The ball is being played towards the right side of the field. The sideline is visible on the right, and the ball appears to be moving in that direction. The trajectory of the ball suggests it is heading towards the sideline. There are no players immediately near the ball to intercept it before it crosses the line. The angle and height of the ball indicate it is likely to continue its path towards the sideline.

**Ground Truth**: B          **Mermaid Flow:**

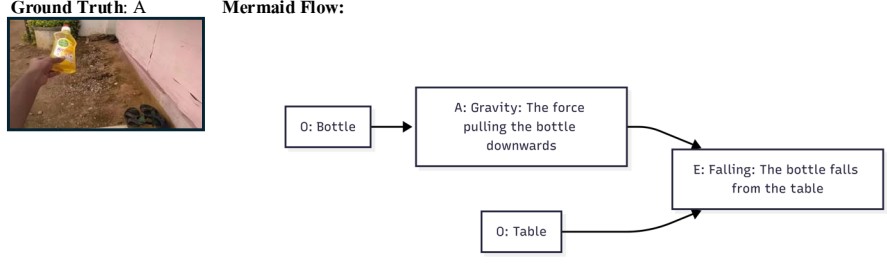

**Question 2:**

If the bottle were pushed gently off the edge of the table, in which direction would it most likely fall?

**Option:**

A. To the ground directly below the table's edge. B. Towards the drainpipe. C. Towards the plant.

**Model Response:** A--The bottle is placed near the edge of a table, and the ground directly below is visible. The drainpipe and the plant are located further away from the table's edge. If the bottle is pushed gently off the edge, it will most likely fall straight down due to gravity, landing directly below the table's edge. The other options are further away and would require additional force or direction to reach.

**Ground Truth**: A          **Mermaid Flow:**

Figure 22: Sample cases of Perception-Scene Reconstruction

Perception: Mechanics Reasoning

**Question 1:**

What is the direct cause of the toothbrush moving from the green t-shirt to the hand?

**Option:**

A. The purple shirt being placed on top of the green t-shirt.     B. The hand picking up the toothbrush.

C. The toothbrush resting on the green t-shirt.

**Model Response:** B--The sequence of images shows a toothbrush initially resting on a green t-shirt. In the third image, a hand appears near the toothbrush. In the fourth image, the hand is holding the toothbrush, lifting it from the t-shirt. By the fifth image, the toothbrush is clearly in the hand. The direct cause of the toothbrush moving from the green t-shirt to the hand is the action of the hand picking it up.

**Ground Truth**: A     **Mermaid Flow:**

**Question 2:**

What is the primary cause of the flip-flop being lifted from the floor?

**Option:**

A: The weight of the flip-flop. B: The pulling force applied by the hand.

C: Friction between the flip-flop and the floor.

**Model Response:** B--The sequence of images shows a hand interacting with a pair of flip-flops on the floor. Initially, the flip-flops are stationary on the floor. As the hand makes contact and begins to pull, the flip-flops start to lift from the floor. The consistent presence of the hand pulling the flip-flops in each frame indicates that the primary cause of the flip-flops being lifted is the pulling force applied by the hand. There is no indication that the weight of the flip-flops or friction is causing the lift.

**Ground Truth**: B     **Mermaid Flow:**

Figure 23: Sample cases of Perception-Mechanics Reasoning

were entirely conceived and conducted by the authors. The LLM did not contribute to the design of the framework, the experimental results, or the interpretation of findings. All scientific content remains the responsibility of the authors.

