# OpenReview forum: "Causal Scaffolding for Physical Reasoning: A Benchmark for Causally-Informed Physical World Understanding in VLMs"
_ICLR.cc/2026/Conference — ICLR 2026 Conference Withdrawn Submission_

### Official Review · Reviewer_NtqG · 2025-10-29

**Soundness:** 3
**Presentation:** 2
**Contribution:** 2
**Rating:** 4
**Confidence:** 5

**Summary:**

This paper introduces CausalPhys, a novel benchmark designed to evaluate and improve the causal physical reasoning capabilities of Vision-Language Models (VLMs). The benchmark includes over 3,000 video and image-based questions across four domains—Perception, Anticipation, Intervention, and Goal Orientation—each annotated with a ground-truth causal graph. The authors also propose a causal-graph-grounded evaluation framework and a fine-tuning strategy called Causal Rationale Fine-Tuning (CRFT) to enhance model reasoning. Extensive experiments reveal that current VLMs struggle with causal reasoning, especially in capturing relational dependencies, and demonstrate that CRFT improves both accuracy and interpretability.

**Strengths:**

1)	CausalPhys is the first benchmark to provide explicit causal graph annotations for physical reasoning tasks, enabling fine-grained, mechanism-level evaluation.
2)	The taxonomy aligns with Pearl’s causal hierarchy, offering a principled framework for diagnosing model capabilities across different levels of causal reasoning.

**Weaknesses:**

1)	Although the authors claim that they proposed a Causal Rationale informed Fine-Tuning strategy (CRFT) that scaffolds VLM reasoning with causal graphs., I cannot find the CRFT details in the paper.

2)	The paper does not fully detail the process or inter-annotator agreement for causal graph construction. Is the causal graph annotated automatically or by humans? More transparency on annotation guidelines and quality control would enhance credibility.

3)	Using an LLM to evaluate causal faithfulness may introduce bias or inconsistency. A human evaluation or more robust automated metric (e.g., graph alignment scoring) could complement the current approach.

4)	While CRFT shows promising results on Qwen2-VL, its effectiveness across a wider range of model architectures and scales remains to be validated.

5)	The paper would benefit from ablations on the contribution of different causal graph components (objects, attributes, events) and the impact of rationale quality on CRFT performance.

**Questions:**

See the weakness.

---

### Official Review · Reviewer_pW9i · 2025-10-31

**Soundness:** 1
**Presentation:** 1
**Contribution:** 2
**Rating:** 2
**Confidence:** 4

**Summary:**

The paper studies the problem of *physical commonsense reasoning* in vision-language models (VLMs). It makes three main contributions: (1) the introduction of **CausalPhys**, a benchmark of multiple-choice visual QA instances, where each instance is accompanied by a structured causal graph that represents the underlying physical mechanisms in the image or video. All instances, including questions, answers, and causal graphs, are produced through multiple rounds of human annotation. (2) The development of a set of causal-graph-grounded metrics to evaluate model outputs in more depth, moving beyond simple answer accuracy. These metrics include *entity faithfulness* (whether all entities from the causal graph are mentioned in the rationale), *description correctness* (whether entities are described accurately), and *relation awareness* (whether the order of entities in the rationale aligns with the causal graph). (3) The paper also proposes a causally inspired fine-tuning (CRFT) approach, which leverages causal graphs to improve model performance. In short, CRFT is a weighted supervised fine-tuning method that combines gold rationales, derived from converting graph edges into sentences, with the final answers.

The main results in Table 1 evaluate several open- and closed-source models of different sizes on CausalPhys, showing that current models perform poorly on the task and often approach random-guess levels. Table 4 focuses on the CRFT method and shows that Qwen2-VL 7B performs better on the task after applying CRFT compared to standard fine-tuning.

**Strengths:**

Overall, the paper addresses a relevant and challenging problem. Understanding physical reasoning remains an open question for VLMs, and this work makes a useful attempt to explore it in a structured way. The paper is clearly written and includes illustrative examples both in the main text and the appendix.

**Weaknesses:**

the overall execution is weak, and the paper lacks the technical depth and clarity needed for a top-tier venue like ICLR.
L1) The main concern is the lack of detail about the data creation pipeline. Since CausalPhys is the central contribution, a more thorough explanation of how the data were collected and processed would make the paper stronger, currently, only a few lines (LL 256–266) describe this process, which makes it difficult to assess the dataset’s quality or novelty.
L2) Similarly, there is limited information about the construction of the causal graphs or whether their structure is based on prior work.
L3) The reliance on human-supervised annotations in CRFT also raises some questions about scalability and generalizability, which are not addressed. It would be helpful to include discussion or evidence about how CRFT might transfer to out-of-domain scenarios.
L4) Finally, the evaluation section could use more clarity and consistency. For instance, the baseline results for Qwen2-VL 7B differ between Tables 1 and 4 (ACC:  0.5349 vs 0.5065), and more information about the experimental setup (e.g., train–test splits, benchmark divisions) would make the results easier to interpret.

**Questions:**

N/A

**Details Of Ethics Concerns:**

The paper omits an ethical discussion, which is important given the reliance on human annotators. Details such as selection criteria, qualifications, compensation, and potential exposure to sensitive or offensive content are not provided. These are essential considerations for any dataset-based contribution.

---

### Official Review · Reviewer_YqdB · 2025-11-02

**Soundness:** 3
**Presentation:** 3
**Contribution:** 2
**Rating:** 4
**Confidence:** 3

**Summary:**

This paper introduces CausalPhys, a ~3k-item image/video benchmark for physical reasoning organized along Pearl’s ladder (Perception, Anticipation, Intervention, Goal Orientation), where each question is paired with an instance‑specific causal graph (objects, attributes, events). Beyond answer accuracy, the authors propose three rationale-based metrics—Entity Faithfulness (EF), Description Correctness (DC), and Relation Awareness (RA)—scored by an LLM-as-judge. They also present Causal Rationale Fine‑Tuning (CRFT), which supervises models with gold rationales aligned to the ground‑truth causal graphs. Experiments across open/closed VLMs show moderate accuracy but poor relation reasoning; CRFT reportedly improves interpretability metrics with competitive accuracy. Key elements are in Fig. 1 (p. 2) and Table 1 (p. 3) for the taxonomy and dataset stats; Table 3 (p. 6) summarizes benchmark results; Fig. 4 (p. 7) details the evaluation workflow; Fig. 5 (p. 8) and Table 4 (p. 9) present CRFT.

**Strengths:**

1.	Well-scoped problem & framing. Grounding tasks in causal rungs and annotating per‑instance DAGs is thoughtful and aligns with the community’s push toward mechanistic evaluation rather than answer-only scoring (Fig. 1, Table 1).
2.	Metricization of rationales. EF/DC/RA provide a structured view of “how” models reason, not just “what” they answer (Sec. 3.2–3.3; Fig. 4).
3.	Transparency gestures. The paper promises to release the data construction pipeline, Mermaid DAGs, JSON schema/validators, and prompts (Appendix, pp. 14–17), which—if delivered—aid reproducibility.
4.	Empirical sweep. The comparison includes several open and closed VLMs; the observation that relation reasoning lags accuracy is consistently supported by Table 3 (RA ≈0.20–0.31 while ACC ≈0.35–0.60).

**Weaknesses:**

1. Heavy reliance on LLM‑as‑judge, limited validation. EF/DC/RA are adjudicated by a single LLM judge with True/False questions (Appendix prompt, p. 17). There is no analysis of inter‑judge agreement, cross‑LLM robustness, calibration, or sensitivity to prompt paraphrases. This is critical because conclusions (e.g., “CRFT improves interpretability”) stand on these scores.

(Personal suggestion, since LLM-as-judge is not reliable, why not use it as a weak supervision instead of the ground truth?)

2. Benchmark scale and coverage.
~3,000 items across 16 subcategories means some buckets are small (e.g., certain Intervention/Viewpoint cases; see Table 1). This limits per‑subset power, and Table 3 already shows some accuracies near chance, raising concerns about noise and confidence intervals. No uncertainty estimates are reported.

3. The dataset is curated from many public sources (Ethics, p. 10). While common, training exposure for large VLMs and for the teacher LLM that produces gold rationales (CRFT, p. 8) is not probed. There is no explicit de‑duplication/near‑duplicate analysis, data source licensing table, or “seen vs unseen” test to contextualize scores.

4. While CausalPhys introduces per-instance causal graphs and rationale-based evaluation, its overall task setup—multimodal multiple-choice physical reasoning over four domains of perception, relationships, scene understanding, and dynamics—closely parallels PhysBench (Chow et al., 2025)

Minor:
Some notation and metric definitions would benefit from sharper formalism (e.g., how DC’s “semantically consistent” is operationalized). The Reproducibility Statement references “Section ?? / Appendix ??” which appear as placeholders (p. 10).

**Questions:**

1. Do EF/DC/RA correlate with human judgments (e.g., Spearman/Pearson on a human‑rated subset)?
2. What is inter‑judge agreement across two different LLM judges and across paraphrased judging prompts?
3. **Novelty relative to PhysBench**: (1) Have you evaluated CRFT-trained models on PhysBench to show that causal scaffolding improves general physical reasoning rather than just metric alignment on your dataset? (2) Since Table 2 of CausalPhys explicitly lists PhysBench as lacking causal structure, can you clarify whether the data or question design in CausalPhys is otherwise new, or mainly a re-annotation of PhysBench-style items?

---

### Official Review · Reviewer_63vL · 2025-11-05

**Soundness:** 3
**Presentation:** 3
**Contribution:** 3
**Rating:** 6
**Confidence:** 4

**Summary:**

The paper introduces CausalPhys, a 3k+ instance image/video benchmark for physical reasoning. For each image/video, a multiple-choice question is paired with an instance-specific causal graph spanning objects, attributes, and events across four categories aligned with Pearl’s ladder. It proposes causal-graph-grounded metrics for evaluation. It proposes the Causal Rationale Fine-Tuning (CRFT) to train VLMs to generate causal-graph-anchored rationales in addition to answers. Experiments show broad RA weaknesses across strong VLMs and indicate CRFT improves accuracy and causal metrics on Qwen2-VL-7B versus vanilla and answer-only SFT.

**Strengths:**

- Novelty: The dataset has a principled coverage across Pearl’s ladder. The Instance-specific DAGs have typed nodes (object/attribute/event) and diverse causal edges.
- Dataset quality: Transparent construction workflow, double annotation/adjudication, and released pipeline artifacts for reproducibility.
- Significance: Systematic finding that Relation Awareness is the strongest correlate of task success highlights the main bottleneck for VLM physical reasoning.

**Weaknesses:**

- Judge reliability and construct validity: EF/DC/RA rely on LLM-based True/False checks over model rationales; RA treats causal order as “mentioned-before” which can be gamed or insensitive to structure. It'll be good to add human audits and inter-judge agreement.
- Potential leakage: Gold “rationales” used for training come from GPT-4o. If the evaluation judge and its prompts resemble the teacher’s prompting or phrasing, models can learn to trigger the judge rather than reason causally.
- Potential teacher bias: The teacher’s style and inductive biases shape the gold rationales. CRFT may inherit these quirks. Gains could reflect imitation of GPT-4o preferences, not robust causal understanding.
- Lack of generalization test. It'll be great to test on existing causal datasets, e.g., Causal VQA, CausalVLBench

Minor:
- "??" in reproducibility section.

**Questions:**

- Judge calibration: What is the agreement between LLM-judges and human annotators on EF/DC/RA over a stratified sample?
- Backbone ablation: Do CRFT gains hold for InternVL3, Llama-based MLLMs, and Qwen3-VL?
- Free-form vs causal rationales: how does finetuning with free-form CoT rationales (no graph references) perform?

---

### Official Review · Reviewer_JqQQ · 2025-11-07

**Soundness:** 3
**Presentation:** 2
**Contribution:** 3
**Rating:** 4
**Confidence:** 4

**Summary:**

This paper proposes a benchmark, so-called CausalPhys, that covers 3000 carefully curated video and image question-answer pairs to test the abilities of current LLMs about physically causal reasoning, which goes beyond object recognition and surface reasoning. This benchmark targets modeling the perception, anticipation, intervention, and goal orientation. Moreover, the authors propose a metric, a causal-graph-grounded metric, which evaluates the LLMs' foundational causal reasonings. Finally, this paper further introduces an SFT that fine-tunes the model under a carefully curated dataset that focuses on causal rationale-informed fine-tuning.

**Strengths:**

1. The investigation of causal reasoning for the complex physical world sounds like an interesting exploration that moves beyond the "low-level" perception and surface reasoning, but fails to deeply understand the real world.

2. The comprehensive evaluations on various released and closed LLM/MLLMs demonstrate the difficulty that the current LLM/MLLMs pre-trained on observational data lack the internal capabilities to deeply comprehend the physical laws of reality. This paves the way for the researchers to explore and develop more enhanced LLMs/MLLMs to better understand the physical world.

3. This paper spends lots of effort to curate a training set that contains causal rationale-informed CoT knowledge, which will then be utilized to enhance the MLLMs' causal reasonings through SFT.

**Weaknesses:**

1. The sources of the benchmark and training dataset in this paper are relatively unclear.

2. Regarding the comparisons with other benchmarks in Table 2, are there any overlaps with other datasets? What are the main differences, considering the data sources and data diversity coverage?

3. In Table 3, the scale of the proposed benchmark and dataset is limited, which raises concern about the data diversity and distributions covering the real world.

4. In terms of the intermediate reasoning graph nodes of the causal reasoning SFT data, does the model need to also output the predictions matching the relative objects, including the spatial position, their recognition, and relationships?

5. Regarding the training objective at Eq(1), why does this paper only allow the rationale and answer loss to be computed separately? Are there any analyses about these? What if allowing all of them to be computed across the whole generated text? What if training on answers first, then rationale, or rationale first, then answers? Are there any ablations to study this effect on the SFT?

**Questions:**

Please refer to my questions above. While I am still concerned about the scale and data diversity of the proposed benchmark and dataset, though the physical causal reasoning sounds like an interesting exploration, there are also some physics-based benchmarks [1] [2]. Can the authors provide more convincing demonstrations of the differences when compared? Otherwise, this makes the proposed benchmark and datasets showcase lots of overlaps with others.

[1] Seeing is Not Reasoning: MVPBench for Graph-based Evaluation of Multi-path Visual Physical CoT

[2] Physbench: Benchmarking and enhancing vision-language models for physical world understanding

**Details Of Ethics Concerns:**

No.

---

### Note · Authors · 2025-11-14

I have read and agree with the venue's withdrawal policy on behalf of myself and my co-authors.